# Heart Rate Variability in Concussed College Athletes: Follow-Up Study and Biological Sex Differences

**DOI:** 10.3390/brainsci13121669

**Published:** 2023-12-01

**Authors:** Mariane Doucet, Hélène Brisebois, Michelle McKerral

**Affiliations:** 1Departement of Psychology, Université de Montréal, Montreal, QC H3C 3J7, Canada; marianne.doucet@umontreal.ca; 2Centre for Interdisciplinary Research in Rehabilitation (CRIR), Institut Universitaire sur la Réadaptation en Déficience Physique de Montréal (IURDPM), CIUSSS du Centre-Sud-de-l’Île-de-Montréal, Montreal, QC H3S 2J4, Canada; 3Departement of Psychology, Collège Montmorency, Laval, QC H7N 5H9, Canada

**Keywords:** concussion, heart rate variability, athletes, college students, clinical research, biological sex-related differences

## Abstract

Finding reliable biomarkers to assess concussions could play a pivotal role in diagnosis, monitoring, and predicting associated risks. The present study aimed to explore the use of heart rate variability (HRV) in the follow-up of concussions among college athletes and to investigate the relationships between biological sex, symptomatology, and HRV values at baseline and after a concussion. Correlations between measures were also analyzed. A total of 169 (55 females) athletes aged 16 to 22 years old completed baseline testing, and 30 (8 females) concussion cases were followed. Baseline assessment (T1) included psychosocial and psychological questionnaires, symptoms report, and four minutes of HRV recording. In the event of a concussion, athletes underwent re-testing within 72 h (T2) and before returning to play (T3). Baseline findings revealed that girls had higher %VLF while sitting than boys, and a small negligible correlation was identified between %HF and total symptoms score as well as %HF and affective sx. Post-concussion analyses demonstrated a significant effect of time × position × biological sex for %HF, where girls exhibited higher %HF at T3. These findings suggest disruptions in HRV following a concussion and underscore biological sex as an important factor in the analysis of HRV variation in concussion recovery trajectory.

## 1. Introduction

The clinical and scientific communities now recognize concussions as a public health issue, and sports-related concussions are observed with increasing frequency among children and teens who practice contact sports [1,2,3]. It is estimated that each year, between 20 and 40% of athletes suffer a concussion [4]. Although concussions can result in neuropathological changes in some instances (e.g., microhemorrhage, possible diffuse axonal injury), the symptoms are thought to largely reflect functional rather than structural impairment [5,6,7]. Hence, conventional clinical structural imaging techniques cannot usually detect a concussion [5].

A concussion often results in deficits in physical and cognitive areas. There can be vision and balance problems or neck pain that may be related to cervical spine problems, as well as difficulty with physical exertion. Very common are also cognitive difficulties (e.g., attention deficit, memory problems, mood disturbance) that can occur for a variety of reasons such as sleep, visual or vestibular alterations, diminished cognitive capacity, or psychological adjustment challenges [8]. For teens and young adults, signs and symptoms usually resolve within 1 to 3 weeks, but 10–35% of them will need more time to recover [5,9].

Concussions can lead to perturbations in the autonomic nervous system (ANS). Some studies have shown that concussions can affect the balance between the parasympathetic nervous system (PNS) and the sympathetic nervous system (SNS) and therefore lead to fatigue, exercise-induced headaches, and impaired cardiopulmonary response to exercise [10,11,12]. Primarily, excessive activity of the SNS during acute and sub-acute recovery phases has been found, but this is based on few robust studies that have investigated ANS dysfunction related to a concussion [13,14,15,16,17]. It is important to note that some medical conditions with post-concussion-type complaints (e.g., chronic pain, stress, orthopedic injuries) have also been linked to autonomic dysregulation as well as anomalies in cerebral perfusion and baroreflex efficiency [14]. Moreover, anxiety is one of the most frequently diagnosed emotional disturbances following a mild traumatic brain injury, and it is frequently associated with rapid heart rate, shortness of breath, and/or sweating, which are somatic manifestations of the ANS [17,18]. The literature also suggests that the disturbance of the ANS may explain some of the difficulties athletes face when returning to physical activity [13]. A systematic review by Mercier et al. (2022) summarizing the evidence for ANS dysfunction in adults following mild traumatic brain injury found that although ANS dysfunction was often evidenced, the severity and timing of ANS dysfunction remained hardly predictable [19].

In line with best practice protocols for organized sports, when an athlete experiences an impact suspected of causing a concussion, they are removed from the game and assessed with field screening tests [5,20]. Functions such as balance, visual and motor coordination, memory, and language are quickly assessed, as well as self-reported post-concussion symptoms. These field tests are useful for early detection of the presence of a concussion, but their usefulness in determining functional impact and tracking recovery from a concussion is limited [5]. Athletes are then usually followed by a multidisciplinary team, which can include a physiotherapist and a neuropsychologist. Well-established protocols involve post-concussion symptoms reporting and neuropsychological testing when symptoms have resolved [5,21]. An athlete is considered recovered when test performance and symptoms levels are back to a pre-concussion level that is typically assessed via baseline testing [21].

The application of those protocols with student-athletes, especially teens and young adults, poses many challenges. First, prior studies show that 30 to 50% of concussions go unreported [22]. Athletes can choose to not report a concussion for various reasons, mainly because they want to keep playing, they do not want to disappoint teammates or coaches, they do not think a concussion is a serious injury, or they do not recognize the signs and symptoms [23,24,25,26]. Indeed, post-concussion symptoms are non-specific [27]. Furthermore, even when athletes report their concussions, an assessment based solely on symptomatology is problematic, as abnormalities in brain function following a concussion may still be present beyond the point of full symptom recovery [28], as well as physiological problems such as imbalance of the ANS [29]. Finally, neuropsychological testing, such as computerized cognitive assessment, can also pose a challenge, as attempts at “sandbagging” exist, where an athlete purposefully tries to underperform at the baseline assessment to portray a lower estimate of pre-injury ability or performance [30,31].

Objective measures, not based on athletes’ reporting or performance, such as physiological measures, could help to better identify the effects of concussions and follow their recovery so athletes can safely return to play. In addition, physiological recovery trajectory may present differently from clinical recovery trajectory, and emerging research encourages the use of physiological measures to follow a concussion as part of a multimodal approach [32]. To this day, there is no “gold standard” tool to document the trajectory of a concussion. 

Heart rate variability (HRV) is a physiological measure of the fluctuations and variations in the duration of cardiac cycles and measures the time interval between individual heartbeats. It reflects the modulation of the ANS on the cardiovascular system, mainly stress adaptation, and can be used to assess the balance of the PNS and SNS [33,34,35]. The application of HRV in research and clinical settings is growing since it can be measured with minimal hardware and software. According to recent studies, HRV appears to be a promising tool to assess changes to the ANS related to concussions [36,37,38]. 

Standard metrics for measuring HRV are time domain (changes over time) and frequency domain (power spectral density). Recent research suggests that the root mean square of successive differences between normal heartbeats (RMSSD) is the most important time domain metric to look at when monitoring HRV at rest over a short period of time [39]. The existing body of research on HRV and concussions tends to focus more on frequency domain features. Most frequently reported are the VLF, LF, and HF bands. Those frequency bands can be reported either as absolute power (ms^2^) or as relative power (%). VLF, or very low frequency, is the frequency range of 0.0033–0.04 Hz. There is no consensus over what processes VLF truly reflects, but experimental evidence suggests that physical activity and stress modulate its amplitude and frequency [33]. LF, or low frequency, is the frequency range of 0.04–0.15 Hz. It was initially thought to reflect sympathetic activity, but it is now known to be affected by various factors, such as sympathetic and parasympathetic branches, vagally mediated baroreflex, and respiratory rates, making it less valuable for scientific research [40]. On the other hand, HF, or high frequency, refers to the frequency band of 0.15–0.40 Hz. HF is considered a good marker of parasympathetic activity and is highly correlated to RMSSD, as they are both considered to represent PNS activity [41]. 

Several studies have previously demonstrated reproducible differences in HRV measures between healthy subjects and subjects with more severe traumatic brain injury, where the latter showed decreased HRV (mostly decreased total power, LF, and HF) [42,43,44]. Investigating the relationship between HRV and concussions appears to be more complex, as deficits are not as clear, and even more so when HRV is assessed at rest. Indeed, several studies have found differences between subjects with and without concussions when assessing HRV during low-intensity exercise but no differences at rest. Most of these studies showed that across exercise tests, concussed athletes had decreased LF, decreased or increased HF, and/or increased RMSSD [16,40,45,46,47,48]. Other more recent studies have found differences at rest between concussed subjects and controls (concussed participants had decreased HF and increased LF in the acute phase [49] as well as lower pNN50 [11]) or observed variations in HRV when following concussed athletes (decrease in LF between the symptomatic and asymptomatic phases [49]). In two studies, perturbations in HRV were found to persist even after symptom resolution, highlighting the potential for prolonged ANS disruption following a concussion [49,50]. Thus, although HRV appears to be a promising tool to track recovery from a concussion, there is conflicting evidence as to the direction of changes measured and no consensus as to which HRV components are expected to be altered after a concussion [40,51]. A recent systematic review by Flores et al. (2023) concluded that a decrease in HF is expected as the activity of the SNS increases and the activity of the PNS decreases after a concussion [38]. Furthermore, it is important to note that most studies on HRV and concussions included adult participants; the youth were the population of interest in only three studies. In one of those studies, where 29 concussed athletes and 15 controls aged 13 to 18 years old were tested at baseline and during the days following a concussion, no differences on HRV measures between concussed and control participants were found. Concussed participants had increased values of HF and %HF as days post injury increased, and RMSSD decreased after 15 days post injury and continued to decrease until 30 days [10]. This study also found that pNN50, HF, and %HF were higher among those who reported more symptoms [10]. The second study, where concussed adolescents aged 12–18 years old were compared to controls, found reduced RMSSD only in females. The other study, which assessed junior hockey athletes (mean age between 17 and 19 years old), only found differences during exercise [48].

Recent studies suggest that there are biological sex-related differences in concussions, where females suffer more concussions, have greater post-concussive symptoms, and take longer to recover [25,52,53,54,55]. With regard to HRV, in healthy non-athlete adolescents, it is generally found that girls have lower RMSSD, VLF, and LF power but greater HF power [32,56]. In concussed adolescents and young adults, only two studies showed biological sex-related differences, where girls but not boys displayed lower RMSSD 2–6 weeks post concussion [39] or decreased HF power within the first week after injury and until the last measure, which was one week after medical clearance [50]. 

The aim of the present study was to contribute to the improvement of concussion monitoring in college athletes by exploring the clinical utility of HRV, which is a cost-effective, rapid, and non-invasive tool, for measuring the effect of a concussion and tracking its recovery in this population. Our other main objective was to explore biological sex-related differences in this context. Secondary objectives were to investigate symptom variations and correlations between measures at every time point. Our hypotheses were that athletes would show some HRV changes at the post-concussion assessment, and those measures would not necessarily correlate with self-reported symptomatology. Regarding biological sex-related differences, we hypothesized that girls would report more post-concussion symptoms and longer recovery times than boys. No specific hypothesis was put forward regarding specific HRV parameters as there was insufficient converging literature. To our knowledge, except for one other study, which had only one participant [57], our study is the first one to use HRV in a longitudinal design to follow the same college-level athletes across pre- to post-concussion time points and to examine biological sex-related differences in such a design. 

## 2. Materials and Methods

### 2.1. Experimental Design

A prospective longitudinal cohort before-after study design was employed, with three time points for outcome assessment (i.e., preseason baseline—T1, 24 to 72 h post-concussion—T2, and before returning to play—T3). This study is reported according to the STROBE statement for cohort studies [58]. 

### 2.2. Participants

Data were collected from July 2019 to March 2020. A total of 169 student-athletes (55 girls) were included in the study, and 30 (8 girls) of them were diagnosed with concussions for which they were monitored. Table 1 summarizes the number of athletes who agreed to participate in the project by sport and gender. Inclusion criteria were the following: convenience sample consisting of student-athlete participating in the 2019–2020 athletic season, male, female (self-reported biological sex at baseline), ages 16–22 years old. Participants who had a concussion before the start of the season or who were still experiencing symptoms from a previous concussion were excluded from the study. Participation in this research did not entail significant risks or complications. However, since a portion of the assessment was computer-based and required cognitive effort, transient fatigue and headaches were possible. In addition, performance evaluation through the ImPACT test or the administration of mental health questionnaires may have induced some level of anxiety or stress. If this happened, staff was available to intervene and refer the athlete to appropriate resources if needed. As for HRV testing, some discomfort might have arisen during the imposition of a respiratory rhythm for a few minutes, but symptoms quickly returned to normal after a few minutes when athletes resumed their personal respiratory rhythm. Lastly, for post-concussion evaluations, physical symptoms related to a concussion might have occurred during test administration, but breaks and rescheduling were possible. Benefits for participants enrolled in the study were helping to improve the College’s concussion management program and contributing to the advancement of knowledge about concussions in athletes. 

### 2.3. Procedure

Data were collected prospectively as part of a pre-existing concussion management program. At the beginning of the school year, athletes were required to do a baseline test (T1), which consisted of psychological and personal information questionnaires and the computerized assessment tool immediate post-concussion assessment and cognitive testing (ImPACT) [59]. High-risk sports athletes (football and basketball) were required to do an HRV evaluation as well. The HRV test was optional for lower-risk sports athletes (flag football and volleyball). 

A player suspected of sustaining a concussion was identified by either a coach, a sideline physiotherapist, or an athletic trainer based on observed or reported acceleration/deceleration of the head followed by either observed alteration in mental status and/or signs such as confusion and poor balance and/or self-reported symptoms such as headache or nausea [60]. The athlete was removed from play and had to make an appointment with the concussion management team within 72 h post injury. Concussion diagnosis was made by the concussion management team based on history, physical examination (e.g., balance, reaction time), self-reported symptoms, and results on the ImPACT test. The athlete was then re-tested with multiple tests, including HRV (T2). Weekly questionnaires were sent to follow up on the evolution of post-concussion symptoms and psychological health. Clinical decision-making regarding return to school and return to play was made by the physiotherapist and the neuropsychologist, based on the six-stage return to play guidelines issued by the Ministry of Education and Higher Education of the Government of Quebec [60]. At the time of return to play, the athlete was re-tested again with all measures, including HRV (T3). 

### 2.4. Measures

#### 2.4.1. Demographic Information, Psychological Health, and Post-Concussion Symptoms

Age, self-reported biological sex, sport, medical status, and concussion history were collected using an in-house questionnaire. Beck’s Depression Inventory—Second Edition (BDI-II) and Beck’s Anxiety Inventory (BAI) questionnaires were used to measure psychological health. They were selected for this study because of their psychometric qualities (BAI Cronbach’s α: 0.90, BDI-II Cronbach’s α: 0.91) and speed of administration [61,62]. Both questionnaires have a scale that has 21 items rated on a 4-level Likert scale (0 to 3), with higher scores indicating higher levels of depression or anxiety. The respondents were asked to evaluate, for each item, how they had been feeling throughout the previous two weeks. In this study, we used the total score in our analyses. Post-concussion symptoms were evaluated using the symptom rating scale included in the ImPACT test that all athletes took. Participants were asked to report whether in the past 48 h, they had experienced any of 22 post-concussion symptoms, rated on a 7-level Likert scale (none to severe), where higher scores indicate worse functioning [59]. This scale has good internal consistency (Cronbach’s α = 0.88–0.94) [63]. A previous study by Kontos et al. (2012) described four factors to classify post-concussion symptoms: cognitive-sensory (items 1, 12, 13, 19–22), sleep-arousal (items 7, 8, 10, 11), vestibular-somatic (items 2–6), and affective (items 14–17) [64]. These four factors and the total symptoms score were included in the analyses for this study. 

#### 2.4.2. Heart Rate Variability (HRV)

Thought Technology LTD equipment that included a blood volume pulse sensor on the finger and a breath sensor on the chest linked to a ProComp5 infinity encoder were used to assess the HRV. HRV was assessed in two states: two minutes standing (activation of the sympathetic system) and two minutes sitting (decreased sympathetic nervous system activity) [65,66]. These positions were assessed to observe if concussed athletes react appropriately to the change in states and to measure potential variances depending on the position of the athlete. Table 2 presents the HRV variables used in this study based on the literature exposed earlier. 

Artifact correction (e.g., extra or missed heartbeat) was performed with the assistance of the CardioPro Infiniti-Analysis Module VRC-SA7590 software version 1.0 (Thought Technology Ltd., Montreal, QC, Canada), which brings advanced HRV analysis capabilities to the BioGraph platform. HRV data were artifacted following the guidelines suggested by Moss and Shaffer (2019) as well as in the paper by Thought Technology (2010), “Basics of Heart Rate Variability Applied to Psychophysiology” [67,68].

### 2.5. Statistics

The IBM SPSS Statistics version 28.0 for Windows (2021) program was used for all statistical analyses. Separate independent *t*-test (continuous variables) and χ2 analyses (categorical variables) were performed to determine if there were significant group differences between the sexes as well as between participants who sustained a concussion and those who did not. Since the present study included a convenience sample, a priori sample size calculation was not performed [69]. To examine differences over time, mixed model repeated measures ANOVAs with two groups (i.e., boys and girls) and three time points (i.e., preseason [T1], post-concussion [T2], and before returning to play [T3]), were performed within the concussed group. For HRV variables, position (i.e., standing, sitting) was also entered in the equation as a repeated measure. Bonferroni corrections were applied in the post hoc analyses. Estimates of effect sizes were calculated. Pearson correlations between psychological health, post-concussion symptoms, and HRV variables at each time point were computed. 

## 3. Results

### 3.1. Baseline 

#### 3.1.1. Sex-Related Differences

Table 3 presents the summary statistics for biological sex-related differences at baseline. There was a statistically significant difference in age, but it was considered clinically unimportant. No group differences between boys and girls were identified at baseline for presence or number of previous concussions. Girls had a significantly higher score on the BAI, with a large effect size. Girls displayed significantly higher total symptoms score, sleep-arousal symptoms, and affective symptoms, all with medium effect sizes. Regarding HRV values, girls had higher %VLF while sitting than boys, with a large effect size. There were no other significant biological sex-related differences for the various measures at baseline. 

#### 3.1.2. Differences Related to Previous Concussions

Table 4 and Table 5 present summary statistics between athletes who had never sustained a concussion and athletes who had at least one concussion. Looking at differences related to previous concussions, there were no group differences for age nor psychological health. For post-concussion symptoms, there was a significant effect for the vestibular-somatic symptoms factor, with a medium effect size, where athletes who had already sustained at least one concussion in the past exhibited higher scores. There were no statistically significant differences regarding HRV values. 

#### 3.1.3. Correlations between Measures

Table 6 presents correlations between measures at baseline. At baseline, BDI-II and BAI scores were moderately positively correlated together, and they were also positively correlated to all symptom factor scores, with correlation strengths between low and moderate. As it is expected, respective HRV measures (i.e., RMSSD, HF, VLF) were correlated together, with low to moderate correlation sizes. Also, as in previous studies, as RMSSD and HF are thought to both be influenced by the PNS, they were the most strongly correlated together [33,70]. There were small negligible correlations between %HF and total symptoms score as well as %HF and affective sx. 

### 3.2. Post-Concussion 

There were no significant group differences between athletes who sustained a concussion during the sport season and athletes who did not in the number of previous concussions, biological sex, age, BDI score, BAI score, symptoms score, and HRV values at baseline (*p* > 0.05). The average time for return to play was 20.26 (SD = 14.02) days, and there were no differences between boys (M = 21.63, SD = 13.96) and girls (M = 17.14, SD = 14.74), t(21) = 0.682, *p* = 0.509). Apart from two athletes who came in more than one month after their concussion happened, the first post-concussion assessment was 3.46 (SD: 2.35) days after the suspected concussions occurred.

#### 3.2.1. Symptoms 

Figure 1 shows the evolution of each symptom factor and total symptom score across the three time points. Mixed model repeated measures ANOVA showed a significant measure × time interaction for all symptom classes except sleep-arousal. Total symptom score had a significant interaction (F(2, 66) = 3.24, *p* = 0.045), but there were no statistically significant differences between specific times. Cognitive-sensory symptoms (F(2, 67) = 5.080, *p* = 0.009) and affective symptoms (F(2, 68) = 3.63, *p* = 0.032) also showed a significant interaction with time, with a significant difference between T2 and T3 (cognitive symptoms mean difference between T2–T3 = 4.761, *p* = 0.008; affective symptoms mean difference between T2–T3 = 1.723, *p* = 0.031). Another significant interaction with time was with vestibular-somatic symptoms (F(2, 64) = 6.099, *p* = 0.004), with significant differences between T1 and T2 and T2 and T3 and a return to pre-concussion (T1) score at T3 (vestibular-somatic symptoms mean difference between T1–T2 = −1.564, *p* = 0.008; mean difference between T2–T3 = 1.989, *p* = 0.005; mean difference between T1–T3 = 0.425, *p* = 1.000). There was no influence of biological sex on symptom scores. 

#### 3.2.2. HRV Variables 

Table 7 summarizes HRV responses for each position across the three time points. No significant effect of time or biological sex was found for RMSSD, VLF, or HF. A significant effect of time × position × biological sex was detected for %HF (F(4, 78.3) = 3.027, *p* = 0.022). Independent *t*-test revealed that the differences were most important in sitting position. At T3, the difference between boys and girls was of 39% (t(9) = −4.583, *p* = 0.001, d = −3.583). This important difference is illustrated in Figure 2. Independent *t*-test also revealed a significant difference between boys and girls for %VLF while sitting at T3 (t(9) = 2.425, *p* = 0.040, d = 0.952), which is most likely related to the previous result where we found higher %HF for girls at T3. The difference in score for the transition from standing to sitting was not different from the difference found at baseline. Thus, all differences from the transitions were considered normal. 

#### 3.2.3. Correlations between Measures 

At T2, there were no significant correlations between symptoms, biological sex, or presence of previous concussion and VLF, HF, %VLF, and %HF in standing position. There was a moderate negative correlation between RMSSD in standing position and history of concussion (r(25) = −0.527, *p* = 0.005), providing some evidence that athletes who have a history of at least one concussion have lower RMSSD values after having a new concussion, indicating lower PNS activity. Looking at HRV values in sitting position revealed no correlation for RMSSD, VLF, HF, and %HF. A positive moderate correlation between %VLF and baseline BAI score was found (r(16) = 0.550, *p* = 0.018), where athletes with higher BAI scores had higher %VLF right after their concussion. 

At T3, there were no significant correlations between RMSSD, VLF, HF, and %VLF in standing position. There was a negative moderate correlation between %HF and BAI symptoms (r(10) = −0.579, *p* = 0.049), suggesting that BAI decreases as HF increases. This is a result similar and most likely related to what we found between %VLF and BAI at T2. In sitting position, there were no significant correlations for RMSSD, VLF, HF, and %HF. Unsurprisingly, in line with the result illustrated in Figure 2, there was a high positive correlation between %VLF and biological sex (r(9) = 0.837, *p* = 0.001). There were also moderate positive correlations between biological sex and cognitive-sensory symptoms (r(17) = 0.536, *p* = 0.018) and biological sex and vestibular-somatic symptoms (r(17) = 0.623, *p* = 0.004). Finally, affective symptoms were correlated to a history of previous concussion (r(17) = 0.031, *p* = 0.031). 

## 4. Discussion

The aim of this study was to explore HRV trends along the concussion recovery trajectory and to explore biological sex-related differences in measures taken. The secondary objectives included examining variations in symptoms and exploring correlations between measures at each time point. HRV parameters, post-concussion symptoms, and psychological health indicators were examined from baseline to return to play. Findings suggest a different HRV recovery pathway between boys and girls, where for the latter, HRV variables did not appear to go back to baseline level at the return-to-play assessment. No relationship between HRV variables and symptoms was found across concussion recovery, which confirms one of our hypotheses. Regarding symptoms, as expected, girls reported more post-concussion symptoms. Findings also revealed that contrary to what was hypothesized, girls did not have longer recovery time. 

At baseline, our data showed that girls exhibited higher post-concussion symptoms scores than boys. Our findings also indicate significantly higher BAI scores for girls, which is in line with other research [71,72,73]. Our data also indicated a higher BDI score and total symptoms, sleep-arousal Symptoms, and affective symptoms scores in girls, which is also in line with previous findings [72,74,75,76]. One key finding was that girls exhibited higher %VLF in sitting position than boys. This could be associated with higher anxiety, stress, and depressive symptoms in girls, as the VLF band is believed to be associated with efferent SNS activity and its amplitude and frequency modulated by stress responses [33]. In addition, previous studies have found that greater %VLF is related to higher depressive symptoms [39,77,78]. This could also explain why at T2, we found a positive moderate correlation between %VLF and BAI baseline measures, and why we found a negative moderate correlation between %HF at T3 and BAI scores. 

Our analyses on differences between previously concussed athletes and athletes who have never sustained a concussion at baseline finds evidence that previous concussions can lead to more vestibular-somatic symptoms. A similar conclusion was reached by Corwin et al. (2015), where they found that concussed patients who have had three or more previous concussions take longer to recover from their vestibular-ocular abnormalities [79]. Therefore, we speculate that this might mean that vestibular-somatic symptoms can last longer than other symptom classes. Alternatively, it could also be an indicator that vestibular-somatic symptoms are not well detected and therefore tend to go untreated. Indeed, vestibular and oculomotor impairment and symptoms can result in various difficulties in a variety of tasks such as computer work, reading comprehension, and word recognition [80]. These functional impairments can be thought to reflect cognitive deficits rather than vestibular or oculomotor deficits and therefore receive inadequate treatment and persist longer [81].

Moreover, contrary to the study of Blood et al. (2016), we did not find a correlation between HF and depressive symptoms [77], but we did find a small negligible negative correlation between %HF and total symptoms as well as affective symptoms, which could be a milder reflection of the same process. The meta-analytic review by Kemp et al. (2010) also found that HF was negatively correlated with depressive symptoms [82]. A more recent similar meta-analysis by Vreijling et al. (2021) also found that lower HF, reflecting reduced parasympathetic activity, could reflect chronic psychological distress [83]. 

Analyses looking at symptom evolution along the concussion recovery path revealed several interactions between time and measure. This also explains the correlations we found between sex and symptoms measures, where being a girl was an important factor in reporting more symptoms, which agrees with the literature [52,54,74,76,81]. From a clinical point of view, the 9.21 points difference between the post-concussion (T2) total symptoms score and the return to play (T3) score is interesting. Although this shows that athletes report more symptoms right after a concussion than before they return to play, this does not solve the problem regarding self-reported measures. It could be speculated that this might be because athletes were eager to go back to play and therefore reported very few symptoms at T3. A promising finding was that vestibular-somatic symptoms was the only class of symptoms that showed the expected variation (i.e., T1 and T3 scores were similar, T2 score was significantly higher than T1 and T3). This is an important finding, as other studies have found that vestibular and oculomotor symptoms following concussion may be associated with worse outcomes and prolonged recovery times [79,81]. This also suggests that small changes in vestibular-somatic symptoms may be a better indicator of recovery than bigger changes observed in other symptoms clusters.

There were no changes across the time points for RMSSD, VLF, or HF. For %HF while sitting, boys had the expected response, where the PNS is less activated after a concussion. Indeed, the PNS is in charge of rest functions and recovery, and reduced PNS activity is a clear sign of increased stress and poor recovery [84]. A reduction in HF power is also a reflection of the higher sympathetic activation that occurs in the post-acute stage of any injury. In fact, most studies found that HF or %HF went down after a concussion [45,47,49,50]. In our study, it is important to note that for boys, the level of %HF did not go back to baseline level even when they were cleared for play. This result highlights the fact that ANS deficiencies can still be present even when no more symptoms are reported. According to the recent systematic review by Flores et al. (2023), most studies have also found that the resolution of symptoms and the return to play do not necessarily indicate ANS recovery, as HRV measures aren’t necessarily back to normal level at those times [38]. 

Opposite to that is the variation of %HF for girls: there is no difference between the baseline measure and the one right after the concussion; however, at the time of return to play, %HF is much higher. This could indicate that the female PNS is overly activated after a concussion, but this effect is not observable directly after a concussion. Caution should be exercised when interpreting this trend given the limited data set. One other study that did not account for biological sex-related differences also found that %HF values increased as days post injury increased [10]. These results cast a new light on our understanding of the trajectory of HRV in concussions where we can hypothesize a non-linear physiological pattern in recovery, where for some, PNS might increase, and for others, PNS may decrease. Our study suggests that this difference might be explained by biological sex. In line with this result, our results also showed lower %VLF for girls than boys at the time of return to play, hinting that their SNS was less activated. This result was quite surprising, as it was the opposite of what was found at baseline, where girls had higher %VLF than boys. The explanation for this large variation in their SNS activity is still unknown, and it is difficult to explain such results, as there was only one female included at T1 for this measure and only two at T3.

This study shows that HRV could be a potential biomarker to follow ANS disturbance following a concussion and highlights the importance of considering the role of biological sex in clinical conclusions and orientations. However, the exact role HRV could play in concussion diagnosis or as a concussion clearance tool needs further study in larger studies due to the possible delay it takes for HRV changes to appear and to resorb. 

### Limitations

This study has several limitations. Firstly, the COVID-19 pandemic led to school closures and prevented us from retesting all athletes at the end of the athletic season. Therefore, we could not estimate the duration of HRV changes and could not confirm that HRV is stable throughout the year for non-concussed athletes. Future similarly built studies that have baseline assessments for all participants should include a control group that is retested at the end of the study, as physical fitness due to regular sports practice and cognitive fatigue from school could influence HRV and cause it to fluctuate even without a concussion. Nevertheless, previous research has found that HRV tends to remain stable when participants do not undergo major changes, like weight gain [85,86]. Next, our small convenience sample of concussed student-athletes, especially regarding girls, hindered us from conducting more robust analyses on biological sex-related differences, particularly at T3. Despite that, we showed, even in a small sample size, substantial gender distinctions that should be investigated in larger subgroups. Larger, better controlled multisite studies allowing a priori sample size calculations would lead to better powered analyses and more robust interpretations. Finally, ANS regulation and therefore HRV can be influenced by many factors that were not considered in this study, such as the time of day, sleep-wake cycles, and menstrual cycle phase [83,87,88]. Due to our participants being volunteer students, the limited equipment, and time constraint for baseline evaluation, as well as the need for testing to be done as quickly as possible before the start of the season, participants were not able to visit the laboratory on the same day at the same time, nor did they have many time slot options. An assessment of sleep quality, such as a sleep questionnaire, and a question about menstrual cycle phase would be needed in future studies. Similarly, some athletes suffered concussions very early in the season, while others suffered one several months into the season. The school load, fatigue, and psychological consequences of these differences could not be considered, as they are variables we could not control [89]. The physiological mechanisms behind those HRV fluctuations remains unclear, as they could be related to physiological effects of the concussion or psychological stressors of coping with the injury and its recovery. 

## 5. Conclusions

Overall, this prospective longitudinal cohort before-after study suggests disruptions in HRV following a concussion that persist beyond the resolution of symptoms, supporting the idea for prolonged ANS disturbance after a concussion. Biological sex was revealed to be an important factor in the analysis of HRV variation in the concussion recovery trajectory. Because HRV variations do not always appear in the acute phase of a concussion and because it remains unclear at what point HRV goes back to a normal level, this study is not able to support its use for detecting a concussion and using it to clear an athlete. However, future research should investigate in larger cohorts its potential in planning treatment and return to play for college athletes and in tracking ANS recovery. Furthermore, biological sex-related differences should be at the forefront of upcoming investigations. Lastly, clinicians should pay extra attention to vestibular somatic symptoms, as they appear to be the most sensitive to concussions and could last longer than other symptom classes. 

## Figures and Tables

**Figure 1 brainsci-13-01669-f001:**
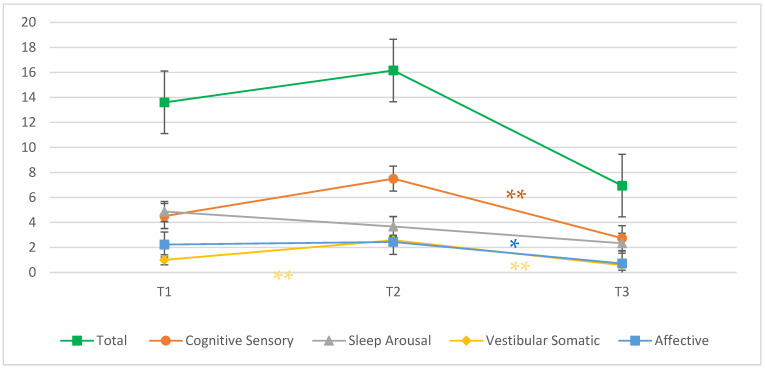
Symptom evolution across three time points; * *p* < 0.05, ** *p* < 0.01.

**Figure 2 brainsci-13-01669-f002:**
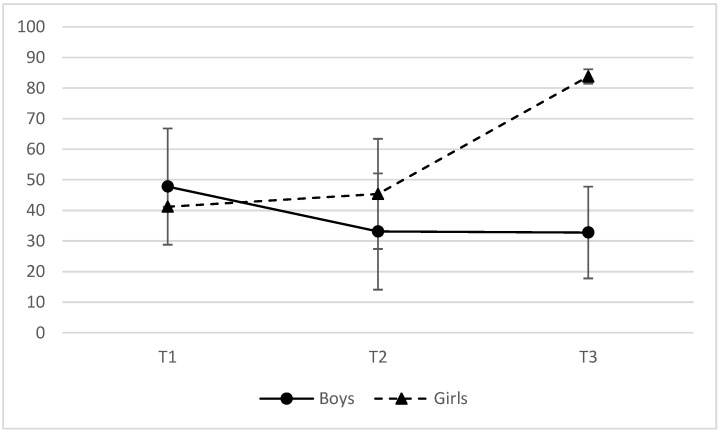
%HF HRV while sitting trajectory across recovery.

**Table 1 brainsci-13-01669-t001:** Characteristics of the student-athletes included in the study.

	Baseline (*n* = 169)	Concussion (*n* = 30)
Age (SD)	18.15 (1.11)	18.17 (1.18)
Biological sex (girls)	114 (55)	22 (8)
Sports (girls)		
Basketball	21 (29)	8 (6)
Volleyball	12 (13)	2 (1)
Football	81 (0)	12 (0)
Flag football	0 (13)	0 (1)

**Table 2 brainsci-13-01669-t002:** HRV variable definitions.

HRV Measure	Units	Definition
RMSSD	ms	Root mean square of successive differences between normal heartbeats
VLF	ms^2^	Absolute power of the very-low-frequency band (0.0033–0.04 Hz)
%VLF	%	Relative power of the very-low-frequency band; [VLF/(HF + LF + VLF)] × 100
HF	ms^2^	Absolute power of high-frequency band (0.15–0.4 Hz)
%HF	%	Relative power of the high-frequency band; [HF/(HF + LF + VLF)] × 100

**Table 3 brainsci-13-01669-t003:** Biological sex-related differences at baseline.

	Boys	Girls	*t*-Test	*p* Value	Cohen’s d
Age	18.32 ± 1.16 (114)	17.82 ± 0.95 (55)	2.978	0.002	0.456
Previous concussion (yes/no)	0.33 ± 0.47 (97)	0.42 ± 0.50 (53)	−1.036	0.151	−0.177
Number of previous concussions	0.45 ± 0.85 (97)	0.70 ± 1.03 (53)	−1.473	0.072	−0.266
BAI	2.14 ± 3.67 (97)	6.53 ± 6.52 (53)	−4.519	≤0.001	0.901 **
BDI-II	5.65 ± 5.92 (98)	7.89 ± 7.16 (53)	−1.940	0.042	−0.350
Total Sx	8.52 ± 9.08 (107)	15.02 ± 11.76 (50)	−3.456	≤0.001	−0.649 *
Cognitive-Sensory Sx	2.70 ± 3.64 (107)	4.08 ± 4.28 (50)	−1.970	0.052	−0.358
Sleep-Arousal Sx	3.51 ± 3.23 (107)	5.84 ± 3.67 (50)	−4.021	≤0.001	−0.689 *
Vestibular-Somatic Sx	0.64 ± 1.33 (107)	1.24 ± 2.50 (50)	−1.585	0.059	−0.334
Affective Sx	1.35 ± 2.58 (107)	3.46 ± 4.05 (50)	−3.383	0.001	−0.677 *
RMSSD	73.67 ± 32.50 (77)	71.56 ± 28.44 (16)	1.401	0.810	0.066
VLF power (ms^2^)	444.90 ± 556.44 (77)	753.76 ± 882 (16)	−1.345	0.196	−0.495
%VLF	20.52 ± 13.33 (77)	33.93 ± 17.47 (15)	−3.364	0.001	−0.952 **
HF power (ms^2^)	609.26 ± 465.35 (77)	575.18 ± 459.14 (16)	0.267	0.790	0.073
%HF	40.13 ± 22.67 (77)	32.40 ± 15.35 (15)	1.264	0.210	0.357

() N for each variable, * medium effect size; ** large effect size; all HRV values reported in Table 3 are the HRV values measured in sitting position.

**Table 4 brainsci-13-01669-t004:** Previous concussion differences at baseline.

	No Concussion (n = 96)	≥1 Previous Concussion (n = 54)	*t*-Test	*p* Value	Cohen’s d
Age	18.02 (1.09)	18.52 (1.09)	−2.687	0.008	−0.457
Sex	1.32 (0.47)	1.41 (0.50)	−1.036	0.302	−0.176
BAI	2.71 (3.79)	4.14 (5.05)	−1.749	0.084	−0.332
BDI-II	5.70 (5.07)	5.71 (5.60)	−0.010	0.992	−0.002
Total Sx	8.99 (8.86)	12.00 (10.26)	−1.800	0.074	−0.321
Cognitive-Sensory Sx	2.70 (3.64)	4.08 (4.28)	−1.970	0.052	−0.358
Sleep-Arousal Sx	4.20 (3.61)	4.37 (3.37)	−0.263	0.793	−0.047
Vestibular-Somatic Sx	0.36 (0.89)	1.08 (1.47)	−3.097	0.003	−0.639 *
Affective Sx	1.43 (2.30)	1.98 (2.73)	−1.238	0.218	−0.223

*: Medium effect size.

**Table 5 brainsci-13-01669-t005:** HRV differences related to previous concussion at baseline.

	No Concussion (n = 58)	≥1 Previous Concussion (n = 29)	*t*-Test	*p* Value	Cohen’s d
RMSSD	74.41 (32.80)	73.16 (31.48)	0.168	0.867	0.039
VLF power (ms^2^)	539.71 (668.80)	341.40 (353.37)	1.788	0.078	0.339
%VLF	24.41 (15.51)	18.97 (12.62)	1.602	0.113	0.372
HF power (ms^2^)	571.75 (445.55)	631.83 (451.61)	−0.582	0.562	−0.134
%HF	38.40 (21.84)	40.85 (22.50)	−0.487	0.627	−0.111

All HRV values reported in Table are the HRV values measured in sitting position.

**Table 6 brainsci-13-01669-t006:** Pearson correlation matrix of baseline measures.

	BAI	BDI-II	Total Sx	Cogn.-Sensory Sx	Sleep-Arousal Sx	Vestibular-Som. Sx	Affective Sx	RMSSD	VLF Power (ms^2^)	%VLF	HF Power (ms^2^)
BDI-II	0.619 **										
Total Sx	0.514 **	0.540 **									
Cogn.-Sensory Sx	0.388 **	0.483 **	0.869 **								
Sleep-Arousal Sx	0.358 **	0.380 **	0.765 **	0.527 **							
Vestibular-Som. Sx	0.370 **	0.345 **	0.656 **	0.499 **	0.282 **						
Affective Sx	0.528 **	0.506 **	0.860 **	0.670 **	0.522 **	0.537 **					
RMSSD	−0.025	−0.005	−0.125	−0.121	−0.033	−0.042	−0.119				
VLF power (ms^2^)	−0.103	−0.005	0.033	0.008	0.095	0.009	−0.024	0.312 **			
%VLF	−0.200	−0.088	0.053	0.047	0.101	−0.056	0.005	−0.020	0.739 **		
HF power (ms^2^)	−0.067	−0.050	−0.013	−0.009	0.088	−0.008	−0.068	0.678 **	0.258 *	−0.103	
%HF	−0.094	−0.017	−0.215 *	−0.048	−0.161	−0.169	−0.271 *	−0.002	−0.418 **	−0.418 **	0.278 **

* *p* < 0.05; ** *p* ≤ 0.001. All HRV values reported in Table 6 are the HRV values measured in sitting position.

**Table 7 brainsci-13-01669-t007:** Concussed participants’ HRV data across different positions and time points.

	T1				T2				T3			
	Standing		Sitting		Standing		Sitting		Standing		Sitting	
	Boys (14)	Girls (1)	Boys (14)	Girls (1)	Boys (19)	Girls (8)	Boys (16)	Girls (6)	Boys (10)	Girls (2)	Boys (9)	Girls (2)
RMSSD	46.05	13.08	76.05	52.34	44.93	41.10	83.94	76.68	38.15	42.88	90.55	116.46
VLF (ms^2^)	322.46	219.20	304.99	122.24	249.44	276.41	429.44	708.65	376.39	354.19	415.73	62.16
%VLF	33.45	71.58	17.69	36.84	21.55	38.90	22.56	33.27	25.65	41.54	22.58	3.79
HF (ms^2^)	395.56	46.60	654.11	136.70	243.07	189.92	897.74	1188.41	173.80	193.01	870.39	1184.58
%HF	24.67	15.22	47.80	41.20	28.85	27.18	33.10	45.41	16.95	26.51	32.78	83.80

## Data Availability

The data presented in this study are available on request from the corresponding author. The data are not publicly available due to privacy reasons.

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
