# Peer review of "Heart Rate Variability in Concussed College Athletes: Follow-Up Study and Biological Sex Differences"

_brainsci, 2023, doi:10.3390/brainsci13121669_

Round 1
Reviewer 1 Report
Comments and Suggestions for Authors
The research on "Heart Rate Variability in Concussed College Athletes: Follow-Up Study and Biological Sex Differences" offers important insights into using HRV to monitor concussions among college athletes.
The study employs a prospective longitudinal cohort design with a diverse range of measurements, including demographic information, psychological health assessments, and HRV evaluations. The main question addressed within the article is the contribution to the improvement of concussion monitoring in college athletes by exploring the clinical utility of Heart Rate Variability (HRV) for measuring the effect of concussion and tracking its recovery, particularly examining biological sex-related differences.
The topic is relevant as it addresses the need for effective concussion monitoring in college athletes. The study explores the clinical utility of HRV, offering a cost-effective, rapid, and non-invasive approach. It is original in its use of HRV in a longitudinal design to follow college-level athletes across pre-post-concussion time points.
This research with HRV analysis adds to the subject area by being one of the first studies to use HRV in a longitudinal design for concussion monitoring in college athletes.
The conclusions seem consistent with the evidence presented. The study highlights disruptions in HRV following a concussion, supporting the idea of prolonged autonomic nervous system disturbance after a concussion.
The references are relevant.
I want to highlight detailed discussion of the study's limitations. Notably, the authors transparently acknowledge the impact of the COVID-19 pandemic on school closures, which disrupted retesting and caution against generalizing HRV stability to non-concussed athletes throughout the year due to these disruptions. The study's limitation section is particularly strong in recognizing the small sample size of concussed student-athletes, especially among females, and its implications for robust analyses.
Tables and figures are of good quality provide all the necessary information to support the study results.
I do not have any methodological concerns about this paper, only ones related to text editting:
1. Restructure a bit the abstract: start with an introductory sentence, then mention: purpose, sample, design, results, practical application.
2. The Introduction can be shorter.
Author Response
- The abstract was restructured, and an introductory sentence was added.
- Some sections of the introduction were removed to make it shorter.

Reviewer 2 Report
Comments and Suggestions for Authors
The current interesting study aimed to explore the use of Heart Rate Variability (HRV) in the follow-up of concussions among college athletes and to establish a framework for interpreting post-concussion changes by investigating the relationships between biological sex, symptomatology, and HRV values at baseline and after a concussion. The small study per se is nice and interesting. However, some revisions regarding a few major and critical comments are needed.
l The introduction needs to reflect more detailed background information about concussion as well as HRV, especially a more detailed description of the effect of concussion on the autonomic nervous system (ANS). If available, it would be extremely valuable to contextualize the experimental findings.
l The sample size for each subgroup is kept to a minimal. The results based on a smaller or moderate sample are mostly unreliable. Consequently, the novelty/innovation of the present study is also greatly diminished. Would there be a minimal sample size needed to avoid underpowered?
l A flowchart describing how you enrolled the participants with the specific inclusion and exclusion criteria in your study was missing in the paper.
l Are there any happened or potential complications resulting from participants joining the study? In contrast, any benefits? These issues need a better description and could be well appreciated. Please briefly mention if any.
l Discussion of the more recent literature on the topic is recommended. Some cited references were outdated and even before millennium.
Comments on the Quality of English Languageminor
Author Response
- A more detailed description of the effect of concussion on the ANS was added. This section also adds recent literature.
- To answer the comment about the minimal sample size needed to avoid underpower, these sentences were added to the manuscript:
“Since the present study included a convenience sample, a priori sample size calculation was not performed [68].”
“Larger, better controlled multisite studies allowing a priori sample size calculations would lead to better powered analyses and more robust interpretations.”
- It was suggested that a flowchart describing the enrollment of participants with the exclusion criteria be added. However, we consider that the following sentences that were slightly restructured are clear enough to describe how participants were enrolled and exclusion criteria, and a flowchart would be unnecessary:
“Inclusion criteria were the following: convenience sample consisting of student-athlete participating in the 2019-2020 athletic season, male, female (self-reported biological sex at baseline), ages 16-22 years old. Participants that had a concussion before the start of the season or that were still experiencing symptoms from a previous concussion were excluded from the study.”
- It was brought to our attention that potential complications and benefits resulting from participants joining the study were missing. We added the following:
“Participation in this research did not entail significant risks or complications. However, since a portion of the assessment was computer-based and required cognitive effort, transient fatigue and headaches were possible. Also, performance evaluation through the ImPACT test or the administration of mental health questionnaires may have induced some level of anxiety or stress. If this happened, staff was available to intervene and refer the athlete to appropriate resources if needed. As for HRV testing, some discomfort might have arisen during the imposition of a respiratory rhythm for a few minutes, but symptoms quickly returned to normal after a few minutes when athletes resume their personal respiratory rhythm. Lastly, for post-concussion evaluations, physical symptoms related to concussion might have occurred during test administration but breaks and rescheduling were possible. Benefits for participants enrolled in the study were helping to improve the College’s concussion management program and contributing to the advancement of knowledge about concussions in athletes.”
- More recent literature was added in the introduction about ANS and concussion. Also, all older references are complemented/replaced with more recent references.
